



# Continuous weekly monitoring of methane emissions from the Permian Basin by inversion of TROPOMI satellite observations

Daniel J. Varon[1,2], Daniel J. Jacob[1], Benjamin Hmiel[3], Ritesh Gautam[3], David R. Lyon[3], Mark Omara[3], Melissa Sulprizio[1], Lu Shen[4], Drew Pendergrass[1], Hannah Nesser[1], Zhen Qu[1], Zachary R. Barkley[5], Natasha L. Miles[5], Scott J. Richardson[5], Kenneth J. Davis[5,6], Sudhanshu Pandey[7], Xiao Lu[8], Alba Lorente[9], Tobias Borsdorff[9], Joannes D. Maasakkers[9], Ilse Aben[9]

[1] School of Engineering and Applied Sciences, Harvard University, Cambridge, United States
[2] GHGSat, Inc., Montréal, H2W 1Y5, Canada
[3] Environmental Defense Fund, Washington DC, United States
[4] Department of Atmospheric and Oceanic Sciences, School of Physics, Peking University, Beijing, China
[5] Department of Meteorology and Atmospheric Science, The Pennsylvania State University, University Park, United States
[6] Earth and Environmental Systems Institute, The Pennsylvania State University, University Park, United States
[7] Jet Propulsion Laboratory, California Institute of Technology, Pasadena, United States
[8] School of Atmospheric Sciences, Sun Yat-sen University, Guangzhou, China
[9] SRON Netherlands Institute for Space Research, Leiden, The Netherlands

*Correspondence to*: Daniel J. Varon (danielvaron@g.harvard.edu)

**Abstract.** We quantify weekly methane emissions at 0.25°×0.3125° (≈25×25 km$^2$) resolution from the Permian Basin, the largest oil production basin in the United States, by inverse analysis of satellite observations from the TROPOspheric Monitoring Instrument (TROPOMI) from May 2018 to October 2020. The mean oil and gas emission from the region (± standard deviation of weekly estimates) was 3.7 ± 0.9 Tg a$^{-1}$, higher than previous TROPOMI inversion estimates that may have used too-low prior emissions or biased background assumptions. We find strong week-to-week variability in emissions superimposed on longer-term trends, and these are consistent with independent inferences of temporal emission variability from tower, aircraft, and multispectral satellite data. New well development and local natural gas spot price were significant drivers of variability in emissions over our study period, but the concurrent 50% increase in oil and gas production was not. The methane intensity (methane emitted per unit of methane gas produced) averaged 4.6% ± 1.3% and steadily decreased over the period from 5–6% in 2018 to 3–4% in 2020. While the decreasing trend suggests improvement in operator practices during the study period, methane emissions from the Permian Basin remained high, with methane intensity an order of magnitude above recent industry targets of <0.2%. Our success in using TROPOMI satellite observations for weekly estimates of emissions from a major oil production basin shows promise for application to near-real-time monitoring in support of climate change mitigation efforts.



## 1 Introduction

Sharp reductions in atmospheric methane emissions are needed to slow near-term climate change (IPCC, 2022). Emissions from the oil and gas industry have been identified as a high priority because implementing new practices can mitigate a significant quantity of emissions at no net cost (IEA, 2022). But the magnitude of oil and gas methane emissions is heavily

disputed. National inventories use bottom-up methods to estimate emissions from infrastructure data, limited device-level measurements, and engineering models. This is difficult because emissions from oil and gas production typically involve thousands of individual point sources, some of which may be individually large (Irakulis-Loixate et al., 2021) and others individually small but accumulating to large regional totals (Omara et al., 2022). Complicating matters is the strong and poorly understood temporal variability of emissions, both in terms of the intermittency of individual point sources (Cusworth et al.,

2021a) and regional fluctuations across oil and gas production basins (Lin et al., 2021; Cusworth et al., 2022). Observations of atmospheric methane have been used extensively for top-down estimates of methane emissions through inversion of atmospheric transport models relating emissions to concentrations (Houweling et al., 2016). Here we show how satellite observations can be used to quantify weekly temporal variability in oil and gas methane emissions from a major production basin (the Permian) over a ~30-month period, and we show that this temporal variability can be explained by specific activity

drivers.

Satellite instruments can sense atmospheric methane from backscattered sunlight in the shortwave infrared (SWIR) to monitor methane emissions (Jacob et al., 2016; Jacob et al., 2022). The TROPOspheric Monitoring Instrument (TROPOMI) launched in 2017 on the Sentinel-5 Precursor (S5P) satellite has been transformative in enabling high-resolution mapping of methane emissions at regional scales (de Gouw et al., 2020). TROPOMI provides daily global coverage at $5.5 \times 7$ km$^2$ nadir

pixel resolution (Hu et al., 2018; Schneising et al., 2019; Lorente et al., 2021). A number of studies have used TROPOMI observations for inverse analyses of emissions at the scale of individual oil and gas basins (Zhang et al., 2020; Schneising et al., 2020; Liu et al., 2021; Shen et al., 2021; 2022; McNorton et al., 2022), but have generally focused on seasonal or annual estimates. Emissions can vary on shorter time scales due to economic drivers (Lyon et al., 2021), accidents (Pandey et al., 2019; Cusworth et al., 2021b), and the lifecycle of facilities (Cardoso-Saldaña & Allen, 2020; 2021; Allen et al., 2022). This

short-term variability is important to understand for interpreting field studies and projecting future trends.

The Permian Basin in Texas and New Mexico is the largest oil-producing basin in the United States, accounting for more than 40% of U.S. oil production (FRBD, 2022). Bottom-up and top-down estimates of methane emissions from the Permian vary widely, from 0.6 Tg a$^{-1}$ as reported in the gridded version of the 2012 U.S. Environmental Protection Agency (EPA) greenhouse gas inventory (GHGI; Maasakkers et al., 2016) extrapolated to 2018 (Shen et al., 2022), to more than 3.7

Tg a$^{-1}$ as inferred by Shen et al. (2022) from 2018–2020 TROPOMI satellite observations. Y. Chen et al. (2022) reported combined emissions of 194 t h$^{-1}$ (1.7 Tg a$^{-1}$) from point sources surveyed aerially across the New Mexico Permian Basin from October 2018 through January 2020, corresponding to 9.4% of the region's gross gas production, a methane intensity more than twice that inferred from basin-wide satellite observations (Zhang et al., 2020; Schneising et al., 2020; Liu et al., 2021;

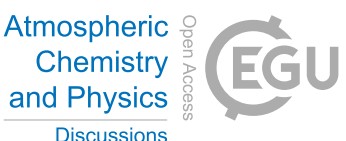

Shen et al., 2022). These discrepancies may be due in part to spatiotemporal emission variability as has been reported for
different sub-regions of the Permian during the 2020 COVID-19 shutdowns (Lyon et al., 2021) and between individual 2019–
2021 aircraft campaigns (Cusworth et al., 2022). High-frequency TROPOMI satellite observations offer a means to probe this
variability at the basin scale.

     Here we conduct an inverse analysis of TROPOMI satellite observations to characterize the magnitude and
spatiotemporal variability of methane emissions from the Permian Basin on a weekly basis, starting from best available bottom-
up prior estimates of emissions and using Bayesian synthesis to obtain optimized posterior estimates assimilating the
information from TROPOMI. We use a Kalman filter to quantify weekly basin-wide emissions at $0.25°\times0.3125°$ ($\approx25\times25$
km$^2$) resolution for 127 weeks from 1 May 2018 to 5 October 2020. The combination of fine spatiotemporal resolution and
long-term basin-wide coverage allows us to directly compare our results with previous space-based and aerial emission
estimates, which cover different time periods and sub-regions within the basin. To evaluate the Kalman filter's ability to
resolve weekly variability, we also compare our results with independent tower and aircraft measurements, inverse analyses
of those measurements, and Sentinel-2 satellite detections of a strong methane point source. Finally, we interpret our weekly
estimates by comparing them with oil and gas activity data and economic variables to gain insight into the factors controlling
methane emissions from the Permian Basin, as a step towards using satellite observations to guide improvements in bottom-
up inventories and identify opportunities to decrease emissions.

**2 Materials and Methods**

**2.1 TROPOMI observations, GEOS-Chem forward model, and prior emission estimates**

TROPOMI was launched into sun-synchronous orbit on the S5P satellite in October 2017. It operates in a nadir-viewing push-
broom configuration with a 2600-km-wide swath and ~13:30 local overpass time (Veefkind et al., 2012). The TROPOMI
operational data record begins in May 2018 and provides daily global observations of dry-air column-average methane mixing
ratios ($X_{CH4}$) at $5.5\times7$ km$^2$ nadir pixel resolution ($7\times7$ km$^2$ before August 2019). Here we use 2.5 years of Version 2.2.0
observations from the scientific data product (Lorente et al., 2021), from 1 May 2018 to 5 October 2020, and select only high-
quality retrievals with quality assurance values greater than 0.5 (Hasekamp et al., 2019). On average we obtain (mean ±
standard deviation) 19346 ± 13073 observations per week over our full inversion domain (96°–110°W, 25°–38°N), including
3062 ± 2314 per week within the Permian itself (Figure 1).

We use the nested version of the GEOS-Chem chemical transport model version 12.7.1 as forward model for the
inversion. The simulation is driven by NASA GEOS Fast Processing (GEOS-FP) meteorological fields at $0.25°\times0.3125°$
resolution over 47 vertical layers from the surface to the mesopause. We use dynamic 3-hour boundary conditions from a
global $4°\times5°$ simulation corrected with spatially and temporally smoothed TROPOMI data as described by Shen et al. (2021).
A one-month spin-up simulation starting from these boundary conditions is used for initialization. TROPOMI observation of
the GEOS-Chem atmosphere is simulated by applying the TROPOMI instrument averaging kernels and prior methane vertical

profiles for individual retrievals to the GEOS-Chem vertical profiles of methane mixing ratios, as described by Varon et al. (2022).

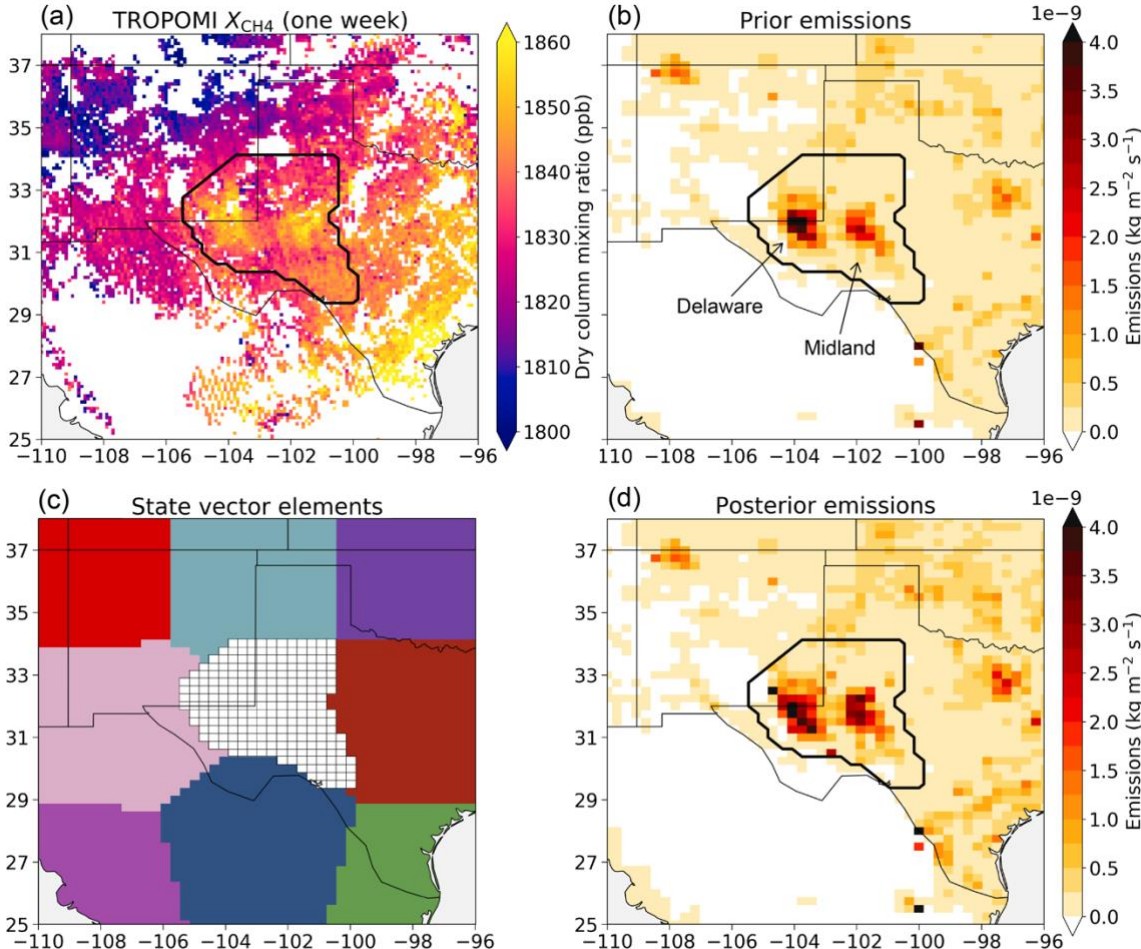

**Figure 1:** TROPOMI methane data, prior emission estimates, and state vector emission elements for our Permian inversion. The thick black contour defines the region of interest for the inversion as the geological extent of the Permian Basin. (a) TROPOMI dry-air column-average methane mixing ratios ($X_{CH4}$) for a typical week (1–8 May 2018), with 23059 observations plotted on a 0.1°×0.1° grid, including 4937 within the Permian. (b) Prior emission estimates on the 0.25°×0.3125° GEOS-Chem forward model grid including the Environmental Defense Fund (EDF) inventory (Zhang et al., 2020) for the Permian. (c) State vector emission elements to be optimized in the inversion, including 235 elements at 0.25°×0.3125° resolution within the Permian and 8 coarse buffer elements to correct boundary conditions. (d) Mean posterior emissions from the weekly inversions for the May 2018 to October 2020 study period, on the same grid as (b).

Our prior emission estimates for the Permian are from the 2018 Environmental Defense Fund (EDF) bottom-up inventory of Zhang et al. 2020 (2.7 Tg a⁻¹; also referred to as the $EI_{ME}$ inventory), which provides a much better representation of atmospheric methane observations than the gridded version of the U.S. EPA GHGI (Maasakkers et al., 2016). The EDF inventory features two emission maxima in the Delaware and Midland sub-basins also seen in the TROPOMI observations (Figure 1). It attributes 94% of Permian emissions to oil and gas activity, and we assume the same fraction for our posterior emission estimates. Prior anthropogenic emissions for the rest of the inversion domain in the U.S. (state vector buffer elements





outside the Permian) are from the gridded EPA GHGI. Anthropogenic emissions for northern Mexico are from the Global Fuel Exploitation Inventory (GFEI; Scarpelli et al., 2020) for oil, gas, and coal emissions, and from the EDGAR v4.3.2 inventory (Janssens-Maenhout et al., 2019) for non-fossil sources. Monthly wetland emissions are from the WetCHARTs v1.2.1

inventory ensemble mean (Bloom et al., 2017), and other natural emissions and sinks are as described by Lu et al. (2022a).

**2.2 Kalman filter inversion methodology**

We quantify weekly average methane emissions from the Permian Basin by Kalman filter inversion of TROPOMI $X_{\mathrm{CH4}}$ observations for 127 weeks from 1 May 2018 to 5 October 2020. In this framework, the posterior emission estimates for each week determine the prior estimates for the next week, enabling the inversion to quantify temporally variable emissions by

providing regular updates as new observations are introduced.

For a given week, the GEOS-Chem forward model $\boldsymbol{F}$ relates the weekly average methane emissions (state vector $\boldsymbol{x}$) to TROPOMI $X_{\mathrm{CH4}}$ observations (observation vector $\boldsymbol{y}$) as $\boldsymbol{y} = \boldsymbol{F}(\boldsymbol{x}) + \boldsymbol{\epsilon_0}$, with observational error $\boldsymbol{\epsilon_0}$ stemming from uncertainties in both the forward model and observations. Here $\boldsymbol{x}$ (length $n = 243$) contains spatially gridded emission elements for the Permian and its surroundings, including 235 elements at 0.25°×0.3125° resolution within the basin and 8

additional coarse-resolution buffer elements adjacent to the basin (Figure 1c) to mitigate boundary-condition errors (Shen et al., 2021; Varon et al., 2022). The observation vector $\boldsymbol{y}$ (length $m$) assembles the TROPOMI observations for the week. The inversion optimizes $\boldsymbol{x}$ to match the observations $\boldsymbol{y}$ subject to constraints from prior information, using optimal estimation by Bayesian synthesis with Gaussian error statistics to retrieve posterior emission estimates $\boldsymbol{x^+}$ of the form (Brasseur and Jacob, 2017)

$$\boldsymbol{x^+} = \mathrm{argmin}_x\,(\boldsymbol{x} - \boldsymbol{x^-})^T \boldsymbol{S^{-1}}(\boldsymbol{x} - \boldsymbol{x^-}) + \gamma(\boldsymbol{y} - \boldsymbol{Kx})^T \boldsymbol{R^{-1}}(\boldsymbol{y} - \boldsymbol{Kx}). \tag{1}$$

Here $\boldsymbol{x^-}$ contains the prior emission estimates; $\boldsymbol{S}$ ($n \times n$) is the prior error covariance matrix describing uncertainty in the prior; $\boldsymbol{R}$ ($m \times m$) is the observational error covariance matrix describing combined uncertainty in the forward model and observations; $\gamma$ is a regularization parameter to prevent overfitting; and the Jacobian matrix $\boldsymbol{K} = \partial\boldsymbol{F}(\boldsymbol{x})/\partial\boldsymbol{x}$ defines the sensitivities of the observations to the emissions as described by the forward model $\boldsymbol{y} = \boldsymbol{F}(\boldsymbol{x})$. $\boldsymbol{F}(\boldsymbol{x})$ is linear so that $\boldsymbol{K}$ fully

defines the forward model for the purpose of the inversion. We compute $\boldsymbol{K}$ column by column via $n$ forward-model simulations perturbing one state vector element at a time. The error covariance matrices $\boldsymbol{S}$ and $\boldsymbol{R}$ are assumed diagonal with uniform error standard deviations of 50% and 15 ppb, respectively. An error standard deviation of 50% with negligible error correlation on the 25 km scale is typical of bottom-up inventories (Maasakkers et al., 2016). An observational error standard deviation of 15 ppb is typical for TROPOMI as determined by the residual error method (Heald et al., 2004; Qu et al., 2021),

and the error correlation is effectively implemented through the regularization factor $\gamma$ (Lu et al., 2021). The standard Kalman filter would allow $\boldsymbol{S}$ to vary from week to week with $\boldsymbol{x^-}$; we discuss the rationale for holding it fixed below. We set $\gamma = 0.25$ following Shen et al. (2021) and investigate the sensitivity to this choice through additional inversions with $\gamma = 0.10$ and $\gamma = 0.30$.





Our Kalman filter to quantify weekly methane emissions involves a prediction step and an update step for each week.

In the prediction step, prior estimates $x_k^-$ of the emissions during week $k$ are made from the most recent posterior estimates $x_{k-1}^+$. Normally this would be done via direct substitution ($x_k^- = x_{k-1}^+$), but we find that this can cause some emission elements to become locked at very low values because the prior error (set as a relative 50%) then becomes very low as well. To avoid this problem, in the prediction step we retain information from the original prior by computing $x_k^-$ as

$$x_k^- = \lambda(\alpha x_{k-1}^+ + [1-\alpha]x_1^-). \tag{2}$$

Here $x_1^-$ contains the prior estimates for the first week (Figure 1b), $\alpha = 0.9$ nudges towards the original prior estimates $x_1^-$ with 90% weight to the latest posterior estimate $x_{k-1}^+$, $\lambda$ is a scale factor to ensure that $x_k^-$ has the same mean as $x_{k-1}^+$, and in the first week ($k = 1$) we have $x_0^+ = x_1^-$. We only use Eq. 2 to compute prior estimates for emissions within the basin; emissions in the external buffer elements (Figure 1) are assigned the same prior estimates (from $x_1^-$) each week. We examine the sensitivity of our results to the choice of $\alpha$ through additional inversions with $\alpha = 0.8$ and $\alpha = 0.95$.

The full Kalman filter prediction step also estimates the prior error covariance matrix $S_k^-$ for the state at time $k$, by applying a state evolution model to the most recent posterior $S_{k-1}^+$. We instead adopt a suboptimal Kalman filter with fixed (diagonal) error covariance matrix $S_k^- = S$ for each week. In the absence of sufficient knowledge to properly define a state evolution model, this prevents the prior error estimates from gradually becoming smaller and weakening further corrections to emissions.

In the update step, the Kalman filter evaluates Eq. 1 for week $k$ to generate posterior estimates of the emissions (Rodgers, 2000)

$$x_k^+ = x_k^- + (\gamma K_k^T R_k^{-1} K_k + S^{-1})^{-1}\gamma K_k^T R_k^{-1}(y_k - K_k x_k^-) \tag{4}$$

and associated error covariance matrix

$$S_k^+ = (\gamma K_k^T R_k^{-1} K_k + S^{-1})^{-1}. \tag{5}$$

The information content of the week's inversion is then computed from the averaging kernel matrix

$$A_k = I_n - S_k^+ S^{-1}, \tag{6}$$

where $I_n$ is the $n$-dimensional identity matrix. The diagonal elements of $A_k$ are called averaging kernel sensitivities and take values between 0 and 1. They describe the extent to which each emission element in the state vector is constrained by the observations of week $k$ as opposed to the prior estimate. An emission element with unit sensitivity is independent of the prior

and entirely constrained by the observations; one with zero sensitivity is not at all constrained by the observations and assumes the prior value. The sum of the averaging kernel sensitivities (trace of $A_k$) defines the degrees of freedom for signal (DOFS). The DOFS value measures the information content of the inversion as the number of independent pieces of information on emissions that can be constrained by the observations.


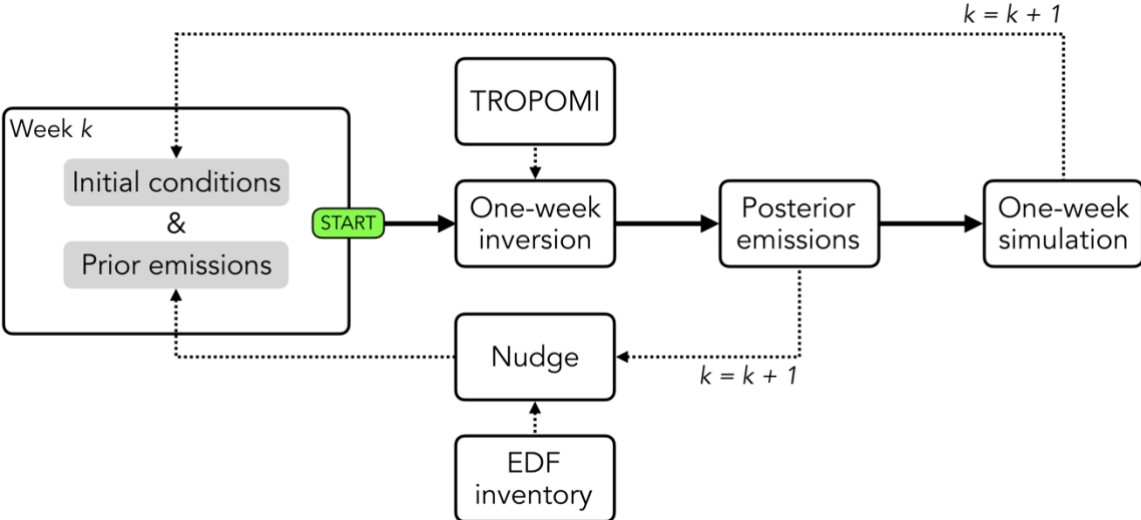

**Figure 2:** Flowchart of our Kalman filter inversion methodology for estimating weekly emissions. Starting from a set of initial conditions and prior emission estimates for the inversion domain (Figure 1), a one-week inversion produces posterior emission estimates for week *k*. The posterior emission estimates are then nudged towards the original EDF inventory emissions before substitution to serve as prior estimates for week *k+1*, and are used in a one-week simulation for week *k* to generate initial conditions for week *k+1*.

Figure 2 illustrates the procedure for advancing the Kalman filter in time. Starting from a set of initial conditions and

prior emission estimates $x_k^-$ for week $k$, a 1-week inversion of TROPOMI observations generates posterior emission estimates $x_k^+$ (Eq. 4). Initial conditions for the first week (1–8 May 2018) are obtained from the one-month spin-up simulation. The posterior estimates are then nudged towards the EDF emission inventory (Figure 1b) following Eq. 2, and the nudged emissions are used as prior estimates for week $k + 1$. Finally, a one-week GEOS-Chem simulation is performed for week $k$, driven by the posterior emission estimates $x_k^+$, to obtain new initial conditions for the following week. This procedure is repeated for the

127 weeks of the May 2018 to October 2020 study period.

## 3 Results and discussion

### 3.1 Permian methane emissions

Figure 3 shows our weekly emission estimates for the Permian region of interest (Figure 1) from 1 May 2018 to 5 October 2020 (127 weeks). Mean weekly emissions for the period expressed as Tg a$^{-1}$ are (mean ± standard deviation) 3.9 ± 1.0 Tg a$^{-1}$

, including 3.7 ± 0.9 Tg a$^{-1}$ from oil and gas. The long-term trend over the 127-week record shows a weak but statistically significant decrease ($p < 0.01$) of –3.5 Gg a$^{-1}$ per week, with a gradual increase of about 20 Gg a$^{-1}$ per week from May 2018 to mid-June 2019 followed by a gradual decrease of about 14 Gg a$^{-1}$ per week through September 2020. Superimposed on these slow trends are strong fluctuations at multi-week scales, often by a factor of 1.5–2 or more, which we examine further in Section 3.6. This includes a rapid reduction in emissions in the months following the March 2020 COVID-19 shutdown, and

several earlier periods of strong variability.



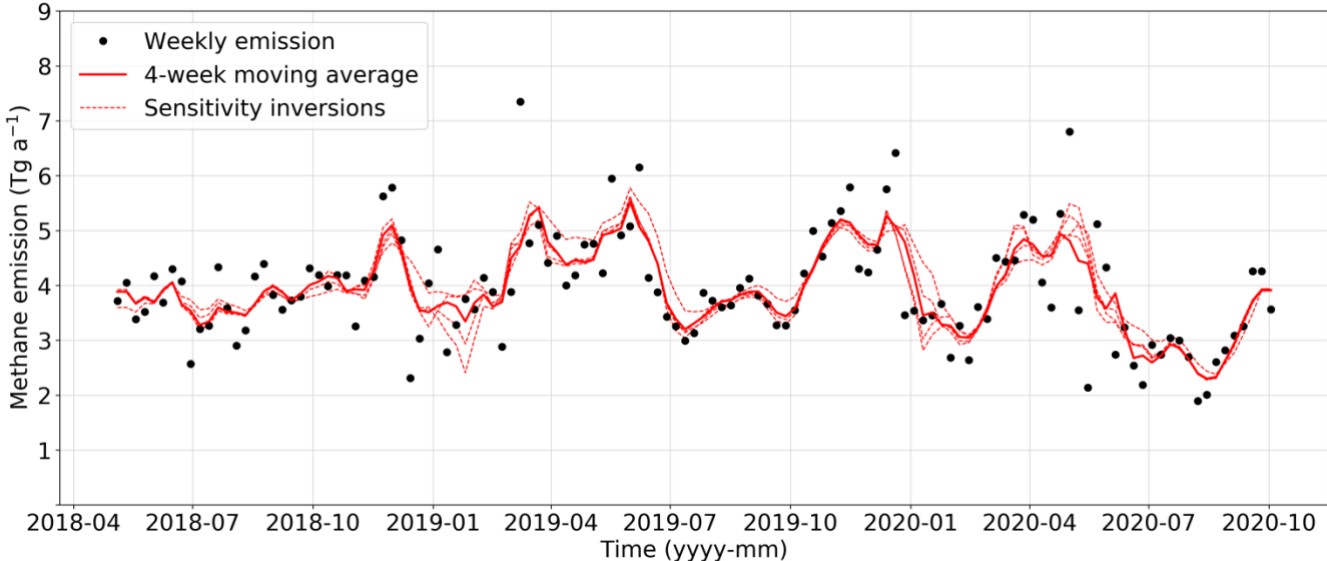

**Figure 3:** Weekly methane emissions from the Permian Basin from 1 May 2018 to 5 October 2020. The Permian is defined by the thick black contour in Figure 1. Emissions are inferred by inversion of weekly TROPOMI satellite observations using a Kalman filter. The mean emission (± standard deviation) is 3.9 ± 1.0 Tg a$^{-1}$, with 3.7 ± 0.9 Tg a$^{-1}$ from oil and gas. The dashed red lines are moving averages of the weekly emission estimates from four sensitivity inversions.


The dashed lines in Figure 3 are from four sensitivity inversions investigating the influence of the parameters $\gamma$ (0.10–0.30) and $\alpha$ (0.80–0.95) on our posterior emission estimates. The posterior mean (3.9–4.1 Tg a$^{-1}$) and standard deviation (0.8–1.0 Tg a$^{-1}$) vary little across these sensitivity inversions. The error standard deviation of the weekly emission estimates, expressed as the standard deviation of residuals between the base inversion and the four sensitivity inversions, is 0.41 Tg a$^{-1}$

(~10% of the mean emission). Increasing $\gamma$ tends to increase the range of weekly emissions by assigning more weight to the observations and thus allowing the optimization more freedom to depart from the prior estimate. Increasing $\alpha$ produces a similar effect by assigning less weight to the EDF bottom-up inventory during the Kalman filter nudging operation (Eq. 2). Overall, the trends we infer in Permian methane emissions are robust to these parameters.

**3.2 Observation density and information content**

Our ability to constrain weekly basin-wide emissions depends on the information content of the Kalman filter as determined by the number of TROPOMI observations and the resulting DOFS. Figure 4 shows how the information content of our weekly inversions varies with the number of TROPOMI observations. Weekly observations for the Permian region of interest (Figure 1) range from 5 to 13288, with a mean of 3062, for our 127 weeks. The resulting weekly DOFS within the Permian range from 0.02 to 14.09, with a mean of 4.67. The number of DOFS increases with observation count by roughly 1.4 DOFS per 1000

TROPOMI observations. The spread around the best-fit line is because the number of DOFS depends also on the absolute uncertainty of the prior emissions, which varies both spatially and temporally, and on the spatial sampling of observations across the inversion domain. Shen et al. (2022) identified DOFS > 0.5 as a practical minimum to estimate total basin methane

emissions with 2σ error ≤ 30% from inverse analysis of TROPOMI observations. Here 124 of our 127 weekly inversions meet the DOFS > 0.5 criterion. TROPOMI thus provides sufficient observational information to support weekly estimation of total

Permian emissions for the vast majority of our study period.

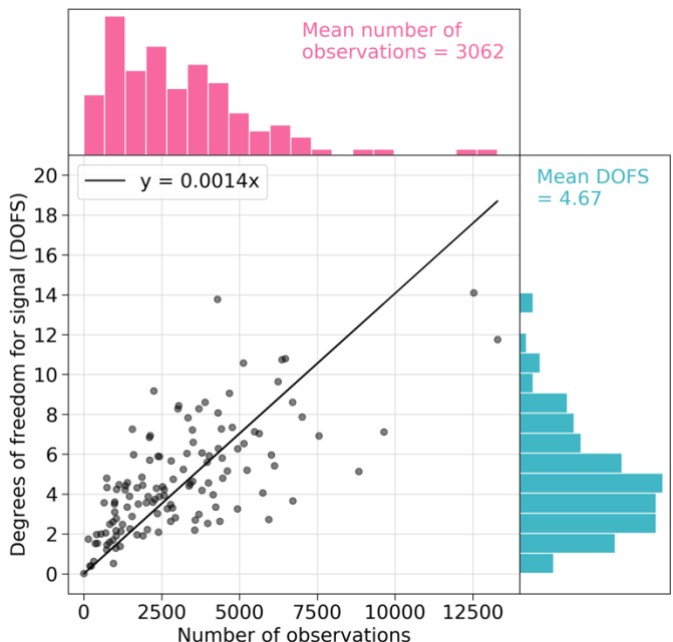

**Figure 4:** Weekly information content of TROPOMI observations over the Permian Basin. The Figure shows the dependence of the weekly degrees of freedom for signal (DOFS) for the inversion on the weekly number of TROPOMI observations in the Permian region of interest (Figure 1), for 127 weeks of inversions. The DOFS represent the number of independent pieces of information on the emission 225 distribution that can be quantified by the inversion independently from the prior estimate. The top histogram shows the distribution of weekly observation counts in the Permian. The right histogram shows the distribution of weekly DOFS in the Permian. Also shown is the reduced-major-axis regression line, indicating 1.4 DOFS per 1000 TROPOMI observations.

### 3.3 Verification with independent tower and aircraft observations

In-situ methane measurements from the Permian Methane Analysis Project (PermianMAP; Lyon et al., 2021) provide an
opportunity for independent evaluation of our results. PermianMAP is an initiative started in November 2019 that gathers methane information from ground-based, airborne, and satellite measurement platforms to monitor emissions from the Permian, with particular attention to a highly active 100 km × 100 km region of the Delaware sub-basin (103.2°–104.2°W, 31.4°–32.4°N). We begin here by comparing our weekly inversions with PermianMAP measurements from the Permian Basin tall tower network (Monteiro et al., 2022). The network comprises five towers, and the three with the most continuous hourly
measurements of methane mole fraction since March 2020 are located near Fort Stockton (30.867°N, 102.815°W), Carlsbad (32.178°N, 104.441°W), and Notrees (31.966°N, 102.770°W), Texas. We compare measurements from these three towers with GEOS-Chem methane simulations driven by (1) constant prior emissions from the EDF bottom-up inventory of Zhang et al. (2020) and (2) our weekly posterior emission estimates.




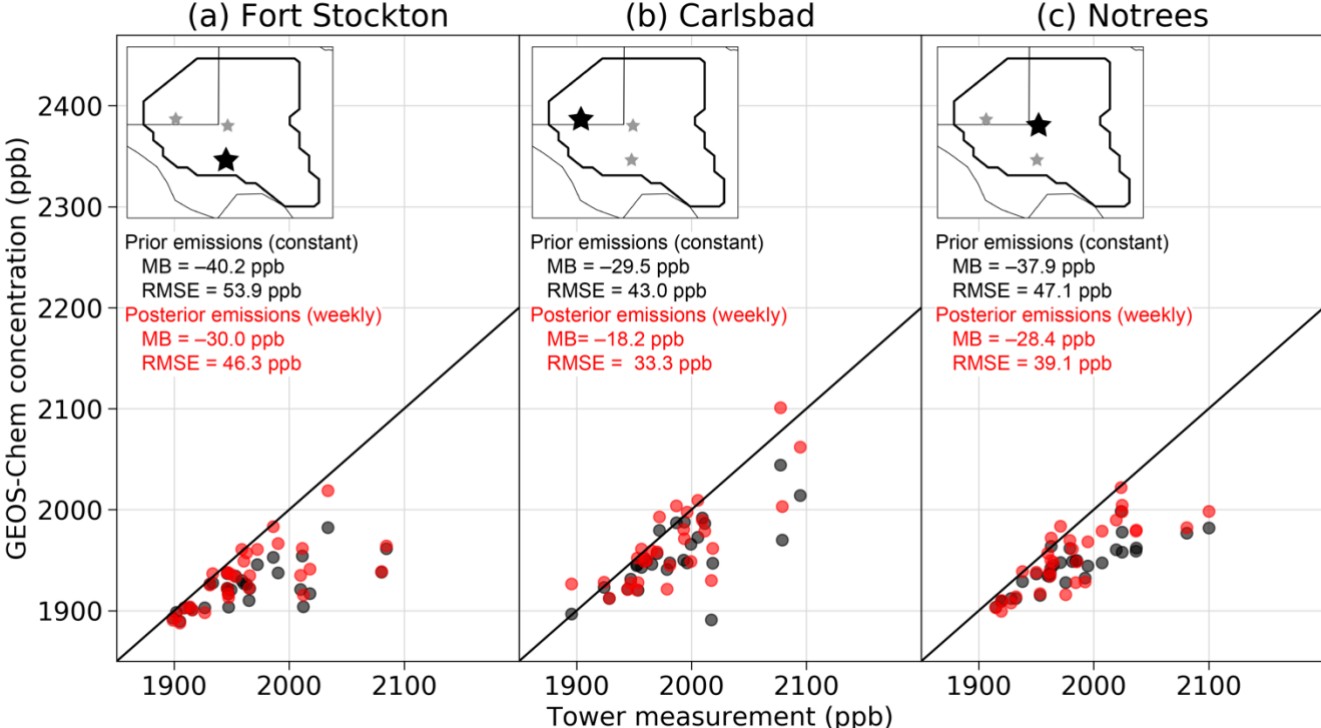

**Figure 5:** Independent evaluation of our weekly methane emission estimates with tall-tower measurements of methane concentrations in the Permian. The figure compares weekly mean afternoon (20:00–24:00 UTC) methane concentrations simulated by GEOS-Chem using prior (constant) and posterior (weekly updated) emissions to tall-tower measurements by Monteiro et al. (2022) from March through September 2020 at Fort Stockton (30.867°N, –102.815°E), Carlsbad (32.178°N, –104.441°E), and Notrees (31.966°N, –102.770°E), Texas. The insets show the locations of the three towers (stars) within the Permian (thick black contour). Also shown are the mean bias (MB) and root-mean square error (RMSE). The 1:1 line is shown in black.

Figure 5 shows the locations of the three towers in the basin and compares their weekly mean methane concentrations with the GEOS-Chem simulations in the nearest model grid cell for afternoon hours (20:00–24:00 UTC) when the mixed layer is fully developed (Barkley et al., 2022). GEOS-Chem underestimates methane concentrations at the three towers, but in all cases our weekly posterior emission estimates improve both the mean bias and RMSE. The improvement in the RMSE is limited by GEOS-Chem representation and transport error, which Lu et al. (2021) previously estimated at 36–68 ppb for surface and tower data.

We further compare in Figure 6 our posterior weekly emissions for the PermianMAP region in January–October 2020 with independent time-resolved estimates from Scientific Aviation aircraft flights on different days and an inverse analysis of measurements from all five towers in the Permian Basin tower network, both reported by Barkley et al. (2022). The satellite and tower inversions generally agree within errors and show a similar decreasing emission trend over the March–October 2020 period. The mean satellite-inferred emission (0.72 Tg a$^{-1}$) is 20% lower than the mean tower (0.88 Tg a$^{-1}$) and Scientific Aviation (0.89 Tg a$^{-1}$) estimates during the period of overlap (March through September 2020). However, linear interpolation





of our weekly inversions yields very close agreement with the individual Scientific Aviation estimates (R = 0.6, RMSE = 0.18 Tg a$^{-1}$, mean deviation = 0.004 Tg a$^{-1}$).

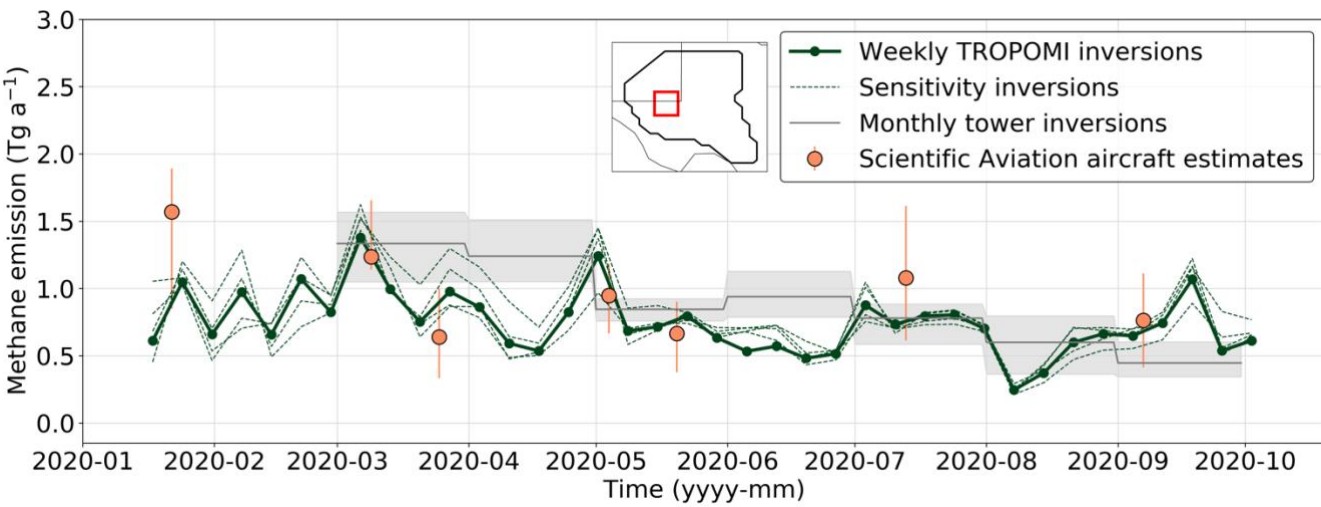


**Figure 6:** Weekly methane emissions from the Lyon et al. (2021) PermianMAP study region (red box in inset: 103.2°–104.2°W, 31.4°–32.4°N). Results from our weekly TROPOMI inversions are compared to monthly tower inversions and aircraft model-based scaling-factor estimates reported by Barkley et al. (2022). The grey shaded area and orange bars represent sensitivity ranges.

**3.4 Comparison with Sentinel-2 point-source observations**

Satellite observations of methane point sources in the Permian provide an additional opportunity to evaluate our inversion results. Single detections of Permian Basin point sources have been reported by Irakulis-Loitxate et al. (2021) and Sánchez-García et al. (2021) using the PRISMA, GF-5, ZY1, and WV-3 satellite instruments. More recently, Ehret et al. (2022) tracked intermittent emissions exceeding ~5 t h$^{-1}$ from a super-emitting compressor station in the Midland sub-basin (31.731°N, 102.042°W) over three months (July–September 2020) using the Sentinel-2 and Landsat-8 satellites. Here we reconstructed

the emissions from that specific point source over the full TROPOMI record by applying the multi-band–multi-pass retrieval method (Varon et al., 2021) to Sentinel-2 observations made over that location every 2–3 days, and quantifying source rates using the integrated mass enhancement (IME) method (Varon et al., 2018; 2021) with local GEOS-FP wind speed data. We obtain a detection threshold of about 2 t h$^{-1}$ as our lowest detected emission.

Figure 7 compares the resulting time series of point source emission rates from the compressor station to the results

of our TROPOMI inversion for the corresponding 0.25°×0.3125° grid cell. The point source was detected on 24 out of 269 non-cloudy passes (9% persistence), with four periods of intermittent activity: May–August 2018, March–May 2019, August–September 2019, and July–September 2020. Our posterior estimates also show emission spikes during these periods, although we do not capture the high values in July–August 2020. Direct comparison between the two time series is challenging because of differences in spatiotemporal scale; there are hundreds of facilities within the 0.25°×0.3125° grid cell, and the Sentinel-2

observations are instantaneous snapshots whereas our TROPOMI analysis averages emissions over the week. Nevertheless,





the mean TROPOMI emissions for that grid cell (9100 kg h$^{-1}$ = 3.1×10$^{-9}$ kg m$^{-2}$ s$^{-1}$) are among the highest in the Permian (Figure 1d).

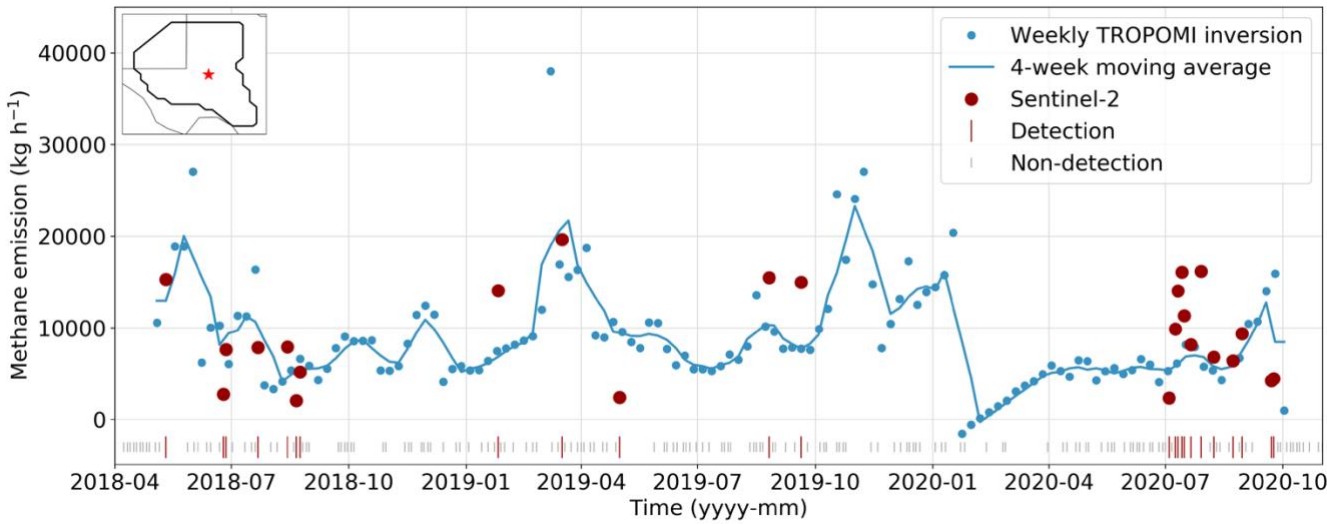

**Figure 7:** Temporal variability of methane emissions from the Permian (Midland) Parks compressor station (31.731°N, 102.042°W; red star in inset map) over the May 2018 to October 2020 period. Point source detections from the Sentinel-2 satellite instrument are compared to weekly TROPOMI estimates for the corresponding 0.25°×0.3125° GEOS-Chem grid cell. The red and grey vertical markers indicate detections and non-detections on non-cloudy passes, respectively.

### 3.5 Comparison with previous TROPOMI estimates of Permian emissions

Table 1 compares our inversion results to previous basin-scale top-down estimates of Permian emissions using TROPOMI observations. The definition of the Permian varies between studies, which may introduce some discrepancies. The studies are also for different periods, but we account for that with our resolved weekly emissions. Zhang et al. (2020) were the first to use TROPOMI to estimate total methane emissions from the basin, and they reported 2.9 ± 0.5 Tg a$^{-1}$ for an 11-month inversion of 2018–2019 data. Schneising et al. (2020) applied a Gaussian integral technique to TROPOMI methane data for 2018–2019 (all 24 months) and estimated total basin emissions of 3.2 ± 1.1 Tg a$^{-1}$. Liu et al. (2021) developed a divergence method for inferring methane emissions from TROPOMI observations and reported 3.1 (2.8, 3.8) Tg a$^{-1}$ for the Permian in 2019. McNorton et al. (2022) obtained total emissions of 2.3 ± 0.5 Tg a$^{-1}$ from a 4D-Variational (4DVar) inversion of TROPOMI data for 15 months during 2019–2020. Shen et al. (2022) reported 3.7 ± 0.5 Tg a$^{-1}$ from oil and gas for a 22-month analytical inversion of TROPOMI observations from May 2018 to February 2020.

Our mean emission for the Permian (3.9 Tg a$^{-1}$; 3.7 Tg a$^{-1}$ for oil and gas only) is 20%–35% higher than these previous TROPOMI-based estimates except for the 3.7 Tg a$^{-1}$ oil and gas emission reported by Shen et al. (2022). In contrast, our emission for the PermianMAP 100 km × 100 km subregion in 2020 is 20% lower than estimated by inverse analysis of tower and aircraft observations (Barkley et al., 2022; Sect. 3.3). The discrepancy between our results and previous (lower) TROPOMI estimates appears to reflect biases in their prior emission estimates and background specification. Zhang et al. (2020) used a





prior estimate of 1.2 Tg a[-1] in their base inversion and included a spatially uniform model bias term in their state vector to
remove the effects of boundary condition (background) errors on the regional inversion. In an alternative inversion with the
EDF inventory (2.7 Tg a[-1]) as prior estimate, they reported 3.2 Tg a[-1]. We use the EDF prior estimate but with TROPOMI-
derived boundary conditions and 8 state-vector buffer elements to guard against boundary-condition error. Shen et al. (2022),
with whom our results agree closely, used a similar prior estimate (2.2 Tg a[-1]) and the same buffer approach but with 16
elements; and when they instead used a 0.6 Tg a[-1] prior estimate, they reported a lower emission of 2.9 Tg a[-1]. Thus the range
of estimates from these three inversion studies can be explained by differences in prior emission estimates and background
specification. McNorton et al. (2022) used a prior estimate of 2.0 Tg a[-1] but obtained only 2.3 Tg a[-1] from a 4DVar inversion
optimizing both emissions and model transport, which could reflect smoothing error if the observations have insufficient
information to constrain both.

Table 1: Comparison of our work with previous TROPOMI estimates of Permian methane emissions. [a]

| Previous study | Study period [b] | This study (Tg a[-1]) [c] | Previous study (Tg a[-1]) [d] | Comments [e] |
|---|---|---|---|---|
| Zhang et al. (2020) | 2018–2019 (11 months) | 4.0 ± 0.9 | 2.9 ± 0.5 | Inversion with scaled 2012 EPA inventory (1.2 Tg a[-1]) as prior emission estimate |
| | | | 3.2 ± 0.5 | Inversion with EDF inventory (2.7 Tg a[-1]) as prior emission estimate |
| Schneising et al. (2020) | 2018–2019 (24 months)[f] | 4.1 ± 0.9 | 3.2 ± 1.1 | Mean result of daily mass balance estimates |
| Liu et al. (2021) | 2019 (12 months) | 4.3 ± 0.9 | 3.1 (2.8, 3.8) | Divergence method |
| McNorton et al. (2022) | 2019–2020 (15 months)[g] | 3.9 ± 1.1 | 2.3 ± 0.5 | 4DVar inversion with prior emission estimate of 2.0 Tg a[-1] |
| Shen et al. (2022)[h] | 2018–2020 (22 months) | 3.8 ± 0.8 | 2.9 ± 0.4 | Inversion with 2018-extrapolated EPA inventory (0.6 Tg a[-1]) as prior emission estimate |
| | | | 3.7 ± 0.5 | Inversion with scaled 2018-extrapolated EPA inventory (2.2 Tg a[-1]) as prior emission estimate |

[a] All authors reported combined emissions for all sectors in the basin, except Shen et al. (2022), who reported oil and gas emissions only.
Oil/gas accounts for 94% of Permian emissions in our work.
[b] Our weekly emission estimates are matched to these different periods, except for Schneising et al. (2020; see footnote).
[c] Values represent mean ± standard deviation of weekly estimates for the period of overlap with the previous study.
[d] The definition of the reported uncertainty varies between studies.
[e] Comments refer to the previous studies.
[f] Includes all 24 months in 2018–2019; we show our period-average result from May 2018 through 2019 for comparison.
[g] Split across two discrete periods.
[h] Shen et al. (2022) performed a 22-month analytical inversion with 16 state vector buffer elements.

The Gaussian integral method of Schneising et al. (2020) and divergence method of Liu et al. (2021) do not rely on

prior estimates but are subject to large errors from uncertainty in the local background subtraction. Schneising et al. (2020)





defined the background from a $2° \times 4°$ upwind box and identified numerous days with negative or very low ($< 1$ Tg a$^{-1}$)
emissions despite stringent filtering criteria for the background observations. Liu et al. (2021) defined the methane background
from the local neighbourhood of individual model grid cells, which aliased Permian enhancements into the background field.

**3.6 Attribution of methane emission trends**

A variety of factors can influence methane emissions from oil and gas activity on weekly to monthly scales, including economic
drivers (Lyon et al., 2021), regulatory changes (Cardoso-Saldaña & Allen, 2021), accidents (Pandey et al., 2019; Cusworth et
al., 2021b), infrastructure development (Lyon et al., 2021), and the lifecycle of facilities (Cardoso-Saldaña & Allen, 2020;
2021; Allen et al., 2022). Figure 8 compares our weekly TROPOMI-derived emission estimates (Figure 8a) with a range of
economic and activity variables (Figure 8b–f) to investigate possible causes of Permian emission variability.

Figure 8b shows monthly data for Permian oil and gas production from Enverus Drillinginfo (Enverus, 2021). Rising
production could lead to higher emissions from increasing activity, but also to lower emissions if more by-produced gas is
brought to market rather than being vented or inefficiently flared. Permian oil and gas production increased steadily by about
50% from May 2018 through March 2020 but this was not associated with an increase in emissions, which may reflect
increased gas marketing. Production declined sharply in April–May 2020 following the COVID-19 shutdown and gradually
recovered from June through September 2020. Lyon et al. (2021) reported concurrent declines in methane emissions and
production levels for the PermianMAP 100 km $\times$ 100 km study region during the COVID-19 shutdown period, but like Barkley
et al. (2022) we find that basin-wide emissions continued to decline after production recovered.

Figure 8c describes monthly trends in new wells entering first production (Enverus, 2021). New well development
can drive variability in emissions because new wells tend to produce high methane emissions in the first few months from
well-completion and tank flashing that then decline rapidly over time (Cardoso-Saldaña & Allen, 2020; Allen et al., 2022).
The number of new wells entering production in the Permian was relatively steady at ~600 new wells per month leading up to
fall 2019, but spiked to ~1200 wells per month in October 2019 and then again in March 2020 before plunging to ~300 wells
per month through September 2020. Both spikes precede a local maximum in emissions by ~1 month, and the sharp reduction
in well development in April 2020 matches the drop in emissions beginning ~1 month later. This confirms the importance of
new wells for driving emissions as was reported by Lyon et al. (2021) for the PermianMAP study region during the 2020
COVID-19 shutdown. If the increases in monthly mean emissions during October 2019 and March 2020 are attributed entirely
to the spikes in new wells, then we estimate 5500–6300 kg d$^{-1}$ per new well during the first month of production. This is 1–2
orders of magnitude higher than bottom-up estimates for new wells in the Barnett Shale and Eagle Ford basins using the
Methane Emission Estimation Tool (MEET; Allen et al., 2022), and may reflect routine flaring and venting of by-produced
gas at new well sites not yet connected to the Permian gas pipeline network (Rystad Energy, 2021). Point sources > 1000 kg
h$^{-1}$ have been frequently detected in previous airborne and satellite surveys of the Permian (Cusworth et al., 2021a; 2022;
Irakulis-Loitxate et al., 2021; Y. Chen et al., 2022).



Figure 8d shows monthly data for Permian natural gas pipeline takeaway capacity and outflows to external markets (S&P Global Platts, 2022). Outflows reflect natural gas production volumes less gas consumed within the basin. Increasing pipeline capacity and outflows could reduce emissions by increasing the marketed proportion of by-produced gas. Pipeline capacity and outflows increased over the study period by 34% and 57%, respectively, in pace with production, and a step

change in capacity occurred in September 2019 when the 2 Bcf d$^{-1}$ Gulf Coast Express (GCX) pipeline entered commercial service (S&P Global Platts, 2019). Paradoxically, in the four months preceding this step change, total Permian outflows appear to have exceeded total capacity. This is likely the result of preparatory (commissioning and line packing) flows in the months leading up to the commercial deployment of the GCX pipeline. Partial capacity up to 2 Bcf d$^{-1}$ of GCX capacity may have become practically available as early as May 2019 (dashed line in Figure 8d), in which case the steep emission reduction we

observe in June 2019, from ~5.5 Tg a$^{-1}$ to ~3 Tg a$^{-1}$, may reflect increased gas marketing facilitated by the new pipeline.

Figure 8e shows the number of monthly flaring targets across the Permian detected by the Visible Infrared Imager Radiometer Suite (VIIRS) satellite instrument using the Elvidge et al. (2016) flare-monitoring methodology (Lyon et al., 2021). Flaring activity produces methane emissions due to incomplete combustion of flared gas (Plant et al., 2022), and unlit flares vent methane directly to the atmosphere. Lyon et al. (2021) reported combustion issues and unlit flares at respectively 6% and

5% of flare sites observed during three Permian surveys performed in 2020. The number of VIIRS-detected flaring targets in the Permian followed an upward trend from May 2018 through August 2019, and then a downward trend through the end of our study period. This is consistent with the gradual increase in emissions seen from May 2018 to mid-June 2019 and subsequent decrease through September 2020, and may reflect increased gas marketing post-GCX. The ~25% spike in emissions observed in November 2018 coincides with a short-lived ~40% increase in flaring targets. The sharply higher flaring

activity during this period may suggest increased methane emissions from flaring.

Figure 8f shows daily spot prices for the West Texas Intermediate Cushing (WTI-Cushing) oil benchmark (EIA, 2022a), the Henry Hub natural gas benchmark (EIA, 2022b), and the local Permian Waha Hub natural gas benchmark (S&P Global Platts, 2022). Trends in oil price could influence methane emissions by motivating more or less activity (e.g., drilling), and trends in natural gas price could change the incentive to market by-produced gas. In contrast with Lyon et al. (2021), we

find that trends in WTI oil price do not track basin-wide emission trends, even during the early-2020 COVID-19 shutdown period. Henry Hub natural gas spot prices correlate positively with increased methane emissions in November 2018 but there is no obvious explanatory mechanism. The steep decline of the Waha Hub natural gas benchmark into negative price territory in March 2019 could explain the concurrent increase in emissions if operators were more motivated to vent or flare by-produced gas, and Figure 8e shows a moderate uptick in flaring activity that month. This dynamic (low Waha Hub price, high flaring,

high emissions) is also present in November 2018, and the sharp May-2019 decline in Waha Hub price matches an increase in emissions as well.

We apply linear regression analysis to the variables in Figure 8 to quantify their influence on emissions. We perform two multiple linear regressions: one using all the variables as predictors to estimate weekly emissions, and another using only the statistically significant predictors ($p < 0.05$) from the first regression. In both cases, the predictors are first resampled to



weekly values and standardized to equal variance (mean = 0, standard deviation = 1). The regressions use Tikhonov ($L_2$)

regularization with leave-one-out cross-validation of the regularization parameter to mitigate collinearity between predictors

(Hastie, Tibshirani, & Friedman, 2009).

summarizes the models and results. The first regression (full model) uses one predictor for combined oil and gas

production (barrels of oil equivalent), two predictors for new well development (number of new wells for the target week, and

number with a four-week lag), and pipeline takeaway capacity assuming that the GCX pipeline began operating in May 2019

(dashed line in Figure 8), for a total of 9 predictors. Only three predictors are statistically significant: new wells (current and

lagged) and Waha Hub natural gas price. New wells correlate positively with emissions and Waha Hub price correlates

negatively. The simplified model uses these three predictors to estimate weekly emissions and shows similar predictive power

to the full model (adjusted $R^2$ = 0.19, from 0.21), but with a much lower Akeike Information Criterion (AIC = −1.12, from

11.04), indicating a better model. Current and lagged new wells remain statistically significant, confirming their importance

as drivers of emissions, but $p$ increases to 0.20 for the Waha Hub natural gas price, suggesting weaker predictive power.

Although the $R^2$ is relatively low, that would be expected because large intermittent point sources such as from the Parks

compressor station of Figure 7 are also likely large contributors to the temporal variability of emissions.

Table 2: Multiple linear (Ridge) regression of weekly Permian methane emissions onto the predictors from Figure 8.

| Model predictor [a] | Full model (9 predictors) [b] | | Simplified model (3 predictors) [c] | |
|---|---|---|---|---|
| | $R^2$ = 0.27    $R^2$ adj. = 0.21    AIC = 11.04 | | $R^2$ = 0.21    $R^2$ adj. = 0.19    AIC = −1.12 | |
| | Coefficient | $p$ | Coefficient | $p$ |
| Intercept (Tg a$^{-1}$) | 3.94 | 0.00 | 3.94 | 0.00 |
| Oil and gas production [d] | −0.0027 | 0.99 | - | - |
| **New wells** [e] | 0.18 | 0.040 | 0.18 | 0.035 |
| **Lagged new wells** [f] | 0.25 | 0.0029 | 0.28 | 0.0011 |
| Pipeline takeaway capacity | 0.050 | 0.83 | - | - |
| Pipeline outflows | −0.16 | 0.65 | - | - |
| VIIRS flaring targets | −0.023 | 0.82 | - | - |
| WTI oil price | −0.11 | 0.35 | - | - |
| Henry Hub gas price | 0.20 | 0.12 | - | - |
| **Waha Hub gas price** | −0.24 | 0.020 | −0.11 | 0.20 |

[a] The predictors are sampled as weekly values and standardized (mean = 0, standard deviation = 1). The model coefficients are thus responses to 1σ increases from the predictor means. Predictors in bold are statistically significant ($p < 0.05$) in the full model regression.
[b] Full model uses 9 predictors representing all the variables of Figure 8, plus an intercept term, to predict weekly emissions.
[c] Simplified model uses only the statistically significant predictors from the full model, plus an intercept term.
[d] Combined monthly production in barrels of oil equivalent.
[e] Monthly new wells for target week.
[f] Monthly new wells with four-week lag.





To sum up, Figure 8 shows that trends in Permian methane emissions reflect multiple factors of variability. New well
development and local natural gas price are likely the most important factors during our study period. Stepping through Figure
8a in reverse chronology: the April 2020 emission peak and subsequent COVID-19 drop were likely caused by trends in new
wells, as were the October 2019 rise and December 2019 decline in emissions. The June-2019 decline may have been the result
of a step change in pipeline takeaway capacity from commissioning and line-packing of the 2 Bcf d$^{-1}$ GCX pipeline, and the
earlier March 2019 rise in emissions may have been caused by plunging Waha Hub natural gas prices. The spike in emissions
observed in late 2018 may reflect low Waha Hub prices and sharply higher flaring activity. Long-term trends beneath this
short-term variability are much weaker, with relatively flat emissions over the study period despite surging production, possibly
due to increased gas marketing.







**Figure 8:** Time series of weekly Permian emission estimates and different oil and gas activity variables. (a) Weekly methane emissions, reproduced from Figure 3. (b) Monthly oil and gas production rates. (c) Monthly number of new wells entering production. (d) Monthly total pipeline outflows and takeaway capacity. The dashed line assumes that the Gulf Coast Express (GCX) pipeline became available for commissioning and line packing flows in May 2019 (see text). (e) Monthly number of VIIRS flaring targets. (f) Daily WTI-Cushing spot prices for oil, and Henry Hub and Waha Hub spot prices for natural gas.



### 3.7 Temporal trend of Permian methane intensity

Figure 9 shows weekly estimates of methane intensity for the Permian, defined as total methane emitted from the oil and gas sector per unit of methane gas produced. The methane intensity is a metric for the amount of methane emitted that could instead have been taken to market, and thus measures the potential for emission reductions. Values in Figure 9 are derived from our posterior emissions (Figure 3) and the monthly production levels of Figure 8b, assuming 80% methane content for Permian natural gas (Alvarez et al., 2018). We find a methane intensity of (mean ± standard deviation) 4.6% ± 1.3%, many times higher

than the < 0.2% intensity target of the industry-based Oil & Gas Climate Initiative based on upstream processes (OGCI 2020). There is an overall decreasing trend of about 0.02% per week or 1% a$^{-1}$, which may reflect increased gas marketing as discussed in Section 3.6. The strong short-term variability we observe in intensity mirrors that in emissions and is consistent with expectations for rapidly evolving oil and gas basins (Cardoso-Saldaña & Allen, 2021).

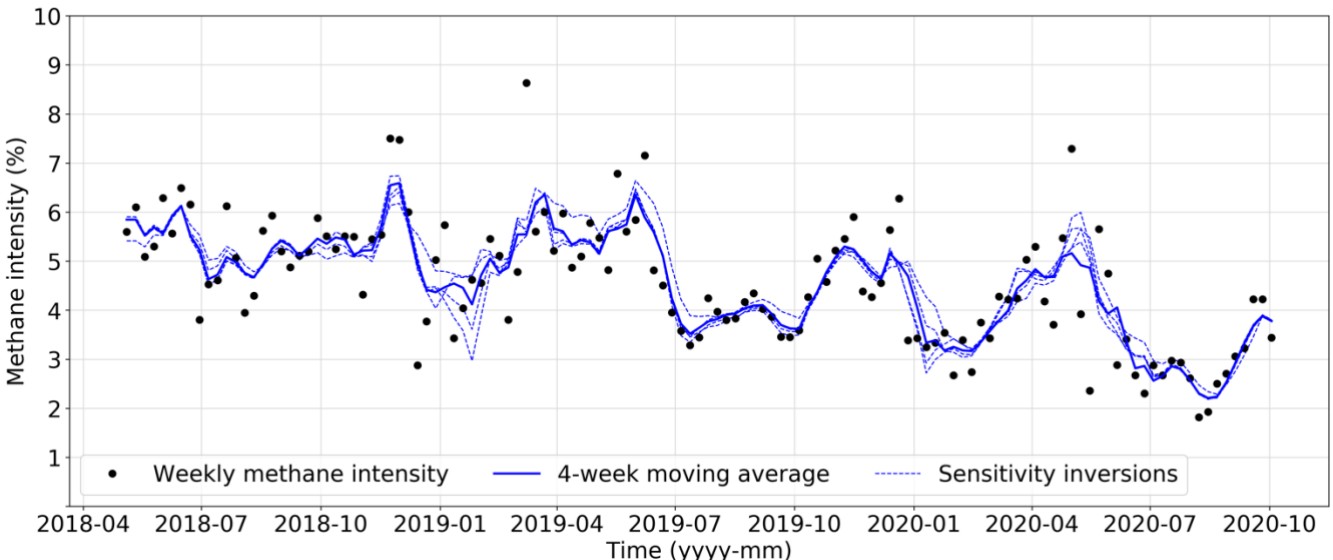

**Figure 9:** Weekly estimates of Permian methane intensity (oil and gas methane emissions divided by methane gas produced), assuming 80% methane content for Permian natural gas. Monthly production is resampled to weekly values for comparison with our emission estimates.

Mean methane intensity in the New Mexico Permian (5.9% ± 2.7%) was higher than in the Texas Permian (4.2% ± 1.2%) during the study period, possibly due to lower production per well in New Mexico (Omara et al., 2022). These intensities are higher than the 3–4% values reported by Lu et al. (2022b) for the Permian in 2018–2019 based on inverse analysis of

surface and GOSAT satellite data, because like some of the previous TROPOMI studies discussed in Section 3.5, those authors used the EPA inventory with very low emissions < 1 Tg a$^{-1}$ as prior estimate. An aerial survey of the New Mexico section of the Permian Basin from October 2018 through January 2020 by Y. Chen et al. (2022) yielded total regional emissions of 194 t h$^{-1}$ (1.7 Tg a$^{-1}$; 1.1–2.3 Tg a$^{-1}$ confidence interval), representing a very high 9.4% (6.1%–12.9%) methane intensity. For the New Mexico Permian during their study period, we find an emission of (mean ± standard deviation) 1.1 ± 0.5 Tg a$^{-1}$ from oil

and gas and a corresponding methane intensity of 6.1% ± 2.8%, 35% lower than their result but consistent within errors.



## 4 Conclusions

We quantified weekly methane emissions from the Permian Basin by Kalman filter inversion of TROPOMI satellite data. Our analysis was performed with the GEOS-Chem chemical transport model at 0.25°×0.3125° resolution for 127 weeks from 1 May 2018 to 5 October 2020. We estimated total emissions of (mean ± standard deviation) 3.9 ± 1.0 Tg a$^{-1}$ over the study period, with 3.7 ± 0.9 Tg a$^{-1}$ from oil and gas, representing a mean methane intensity of 4.6% ± 1.3% (defined as total oil and gas methane emissions per unit of methane gas produced). These levels are nearly four times higher than reported in the gridded version of the U.S. EPA greenhouse gas inventory (GHGI) for 2012, more than six times higher than the extrapolated GHGI for 2018, and an order of magnitude higher than methane intensity targets recently announced by the Oil & Gas Climate Initiative (OGCI). Our mean emission estimate is ~20–35% higher than some previous TROPOMI-based estimates (Schneising et al., 2020; Zhang et al., 2020; Liu et al., 2021), likely due to differences in prior emission estimates and methane background specification, but still within reported uncertainties of those results. For the New Mexico section of the Permian during the Y. Chen et al. (2022) study period we found a mean emission of 1.1 ± 0.5 Tg a$^{-1}$ (6.1% ± 2.8% methane intensity), ~35% lower than their aerial survey reports but consistent within errors and supporting their finding of elevated intensity in that region.

We showed how TROPOMI can quantify both high-frequency and long-term variability in Permian methane emissions. Observations from the Permian Basin tower network, Scientific Aviation aircraft, and the Sentinel-2 satellites provided independent evaluation of our results. Our weekly inversions improved the fit between GEOS-Chem and the tower observations, and reproduced aircraft- and tower-based emission estimates by Barkley et al. (2022) for the PermianMAP 100 km × 100 km study region of the Delaware basin (Lyon et al., 2021). Comparing intermittent Sentinel-2 detections of a compressor station point source with our posterior emission estimates for the corresponding 0.25°×0.3125° grid cell showed roughly consistent variability.

We identified several periods of strong emission variability and investigated their causes by comparing our weekly estimates with oil and gas activity data and economic variables. Periods of rapid growth in emissions were explained by enhanced development of new wells and sharp reductions in local natural gas spot prices. Periods of rapid decline corresponded with reduced new well development (including during the 2020 COVID-19 shutdown) and rapid expansion of Permian pipeline capacity through the Gulf Coast Express project. Linear regression of our weekly emission estimates onto these and other explanatory variables confirmed that new well development and local natural gas price were important (statistically significant) predictors of emission variability during the study period. The sharp emission increases inferred from TROPOMI data in October 2019 and March 2020 following spikes in new well development indicate much higher first-month emissions (5500–6300 kg d$^{-1}$) per new well than previously estimated for the Barnett Shale and Eagle Ford basins using bottom-up methods (Allen et al., 2022).

We found that methane intensity in the Permian decreased over our study period, from 5–6% in 2018 to 3–4% in 2020. This suggests an improvement in operating practices, most apparent in the absence of a large emission increase as

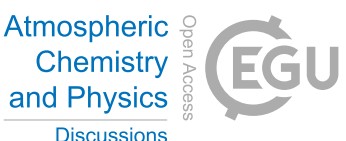

production increased by 50% during the first part of the study period. The methane intensity remains an order of magnitude larger than the 0.2% goal set by industry (OGCI) for oil and gas production by 2025.

490       The recently passed U.S. Inflation Reduction Act includes a methane emissions fee for oil and gas companies and requires U.S. EPA to update the Greenhouse Gas Reporting Program based on measurement data. Our work demonstrates the potential role of satellite observations in validating reported emissions under such policies. This is a step towards near-real-time satellite monitoring of methane emissions as a strategy to evaluate bottom-up inventories, monitor progress on climate goals, and identify sudden increases in emissions. Such monitoring can help improve our understanding of the factors

controlling methane emissions and how emissions respond to economic forcing and policy changes.

## Data availability

The TROPOMI methane data are available at https://ftp.sron.nl/open-access-data-2/TROPOMI/tropomi/ch4/14_14_Lorente_et_al_2020_AMTD/. The Permian Basin tower network data are available at http://dx.doi.org/10.26208/98y5-t941. The Sentinel-2 satellite data are available through the Copernicus Open Access Hub at

https://scihub.copernicus.eu/. Our weekly emission estimates are available at https://doi.org/10.7910/DVN/FBU8NY.

## Author contribution

DJV and DJJ conceptualized the study. DJV, DJJ, MS, LS, DP, HN and ZQ contributed to the methods development. SP performed the Sentinel-2 point source analysis. DJV led the data analysis and interpretation of results with input from all authors. DJV wrote the original draft, and all authors reviewed and edited the manuscript.

## Competing interests

The authors declare that they have no conflict of interest.

## Acknowledgements

This work was supported by the Environmental Defense Fund and by the NASA Carbon Monitoring System. We thank J. Winters and J. Robinson of S&P Global Platts for sharing natural gas throughput and pricing data. A. L. acknowledges funding

from the TROPOMI national program through NSO.





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
