# Peer review of "Continuous weekly monitoring of methane emissions from the Permian Basin by inversion of TROPOMI satellite observations"

_Atmospheric Chemistry and Physics, 2022_

## Referee Comment (RC2)

The MS[ACP-2022-749] by Varon et al., used satellite-based xCH4 observations and inversion modeling to constrain CH4 emissions from oil and gas production hotspot in Permian Basin, the largest oil production basin in the United States at a weekly scale. It's very important to quantify CH4 leakage in these fossil fuel production areas. In general, this MS is easy to follow and well written, considering a few similar studies have been conducted in the same region, some clarification is needed to be highlighted, especially for the most important improvement in this study as displayed below. And it can be accepted after addressing the following comments.

Main comments:

In general, there are a few published papers in the same region that used similar approach (i.e. Zhang et al., 2020), although this submission is conducted at a weekly scale instead of a monthly or annual scale, the authors still need to address the improvement of their study from previous ones, i.e. approach in the inversion framework? or found the weekly relationship between CH4 emissions and other activity indexes? or used more robust observations? or prior emissions and background?

Line 19-21 "The mean oil and gas emission from the region (± standard deviation of weekly estimates) was 3.7 ± 0.9 Tg higher than previous TROPOMI inversion estimates that may have used too-low prior emissions or biased background assumptions". It seems the inversion results are sensitive to prior emissions, have you tested or quantified this potential bias of using different prior emissions to your results?

Line 88-89, "19346 ± 13073 observations per week over our full inversion domain (96°–110°W, 25°–38°N), including 3062 ± 2314", The standard deviation of available data numbers in each week is so high and is comparable with averages, which indicates there are not enough data in some weeks, please address what the potential bias for emission inversion in these weeks with lower available observation numbers. And can you display the time series of available CH4 observation numbers in each week?

For some rainy or cloudy weeks, the available observation data can be sparse, leading to a large missing data gap in the study domain, and how will this situation affect inversion results for this large region? The reason to mention this comment is that your following analysis of the relationship between CH4 emissions and activity indexes ignored the influence of available data.

Line 92-94, "We use dynamic 3-hour boundary conditions from a global 4°×5° simulation corrected with spatially and temporally smoothed TROPOMI data as described by Shen et al. (2021). A one-month spin-up simulation starting from these boundary conditions is used for initialization". As we know that CH4 background uncertainty (bias) will be carried on to calculated CH4 enhancement, which is directly related to posterior CH4 emissions, what the bias of CH4 background in this study and potential uncertainty in deriving CH4 emissions?

Line 110, "It attributes 94% of Permian emissions to oil and gas activity, and we assume the same fraction for our posterior emission estimates.", Can you clarify what is the potential uncertainty of using the constant fraction of 94% to oil and gas activity in inversion results? As I

know most inversion studies have the ability to constrain posterior emissions from different categories.

Section 2.2, As displayed in the reference list, there are some other inversion studies in the same domain, (i.e. Zhang et al., 2020), it's better to illustrate what the main improvement of your study when compared with these previous studies, because it's very hard for audiences to remember and distinguish the method difference between all related studies.

Line 125, "mitigate boundary-condition errors (Shen et al., 2021; Varon et al., 2022).", have you assessed the improvement of CH4 background with observations?

Line 136-137, "The error covariance matrices and are assumed diagonal with uniform error standard deviations of 50% and 15 ppb, respectively", As I understand, the 50% uncertainty for prior inventory may represent regional averages not all grid cells in study domain, which can be much larger than 50%, the same as 15 ppb for observations and GEOS-Chem simulations, so whether the inversion results are sensitive to different values of 50% and 15 ppb, if you assign a slightly larger extent (i.e. 80%, and 20 ppb), how much will the results change?

In figure 3, the weekly emission changes can vary by 100%, indicating the potential bias or uncertainty of CH4 emissions at weekly scale can be much larger than 50%.

Line 190, It seems the use of proportion 94% will largely influence your results of CH4 emissions from oil and gas. I am curious why the inversion model cannot constrain CH4 emissions from each category?

Data displayed in Figure 5 for model simulated xCH4 and observation.

Overall, why tower based CH4 concentrations seem higher than simulations with both prior and posterior emissions (scatter plot is below 1:1 line)? Whether it indicates the posterior CH4 emissions are still underestimated? How about plotting time series of concentration?

Whether one of the reasons for the large difference between model simulation and tower observations is the vertical gradient in the lowest GEOS-Chem model? And what is the height of the lowest model level? or aggregation error for spatial resolution between the point scale and regional scale(25km)?

Line 256-257, "The mean satellite-inferred emission (0.72 Tg is 20% lower than the mean tower (0.88 Tg and Scientific Aviation (0.89 Tg estimates during the period of overlap", From the above concentration comparisons, I also agree that the satellite-inferred emissions are obviously underestimated.

Line 398-410, It's better to display the comparisons between atmospheric inversions and multiple linear regression with figures instead of only using tables and numbers.

Line 438, "assuming 80% methane content for Permian", whether the assumption of using 80% is reasonable, and what the general extent of this value in the study region?

Technical comments

Line 398, "summarizes the models and results" I just guess the first author forget to delete this sentence of comment from other co-authors.

---

## Author Comment (AC1)

**Responses to reviewers**

We thank the reviewers for their comments and suggestions. Each of their comments is copied below in blue, and our responses follow in black. Line numbers refer to the tracked-changes document.

**Reviewer #1**

 The authors describe the inversion approach as using a Kalman filter, but it is unclear to me in what way is this a Kalman filter. On page 6, lines 156-157, they mention that instead of using a "full Kalman filter" they used a "suboptimal Kalman filter with fixed (diagonal) error covariance matrix." If the covariance is fixed, and the model dynamics is not used to update the state nor the errors, how is the scheme a Kalman filter? It is unclear to me what is the difference between this scheme and what is referred to as a Bayesian synthesis inversion? I would describe this as a time-dependent Bayesian synthesis inversion.

Thank you for raising this point. Our algorithm is a suboptimal Kalman filter that assumes a persistence model for state evolution. It is a form of Bayesian synthesis. It is classified as "suboptimal" because it uses a fixed error covariance matrix. To clarify this point, we now include a reference to Todling and Cohn (1994), who discussed suboptimal Kalman filters for atmospheric data assimilation (L. 77), and add the "suboptimal" qualifier on lines 77, 176, 236, 719. Our persistence state evolution model does not seek to predict the dynamics of Permian emissions, because the dynamics are too poorly understood. We now clarify this in Section 2.2 (L. 165, 177-178).

2. Page 3, lines 70-72: The text states that "...starting from best available bottom-up prior estimates of emissions and using Bayesian synthesis to obtain optimized posterior estimates assimilating the information from TROPOMI. We use a Kalman filter to quantify weekly basin-wide emissions." Based on this description, it sounds as though both the Bayesian synthesis inversion and the Kalman filter schemes are being used. It would be helpful if the authors could better explain this. Perhaps the distinction regarding how the two schemes are used can be added to the schematic in Figure 2?

We only use one scheme, the suboptimal Kalman filter with persistence model, which is a form of Bayesian synthesis. To prevent confusion, we replace "synthesis" with "inverse modeling" in the introduction (L. 76). We retain "synthesis" in Section 2.2 documenting our Kalman filter methodology.

**3.** Page 5, line 126: The text states that the observation vector assembles the observations for the week. Are the observations ingested sequentially during the week or does the inversion ingest weekly mean observations? This is somewhat unclear.

The observations are ingested sequentially, not as averages. We now state this explicitly on L. 137-138.

4. Table 2: The discussion in Section 3.5 is focused on comparing the estimated Permian emissions with previous TROPOMI-based estimates (which are given in Table 2). Are the differences in the emission estimates that are compared in Table 2 really meaningful? It seems to me that the errors for these emission estimates all overlap, with the exception of the McNorton et al. (2022) results. Thus, I am not sure how to interpret the discussion in Section 3.5.

Good point! Many of the estimates in Table 1 are consistent within errors and we need to state that. We now do so on L. 378-379. Our discussion of potential biases is meant to address the fact that most previous estimates have central values near 3 Tg  $a^{-1}$ , 20-35% lower than ours.

5. Page 13, lines 309-311: The text here states that the range of reported estimates between this study and the Zhang et al. and Shen et al. studies can be explained by differences in the prior emissions and the background specification. Since this study has produced averaging kernels, the authors could consider substituting the EDF prior emissions with those used in the Zhang et al. and Shen et al. studies to see how much they impact the posterior emissions. The posterior emissions are given by x+ = Ax + (I - A)x-, with the contribution from the prior given by (I - A)x-, where x- is the prior and A is the averaging kernel matrix. How does this contribution change when the other priors are used?

This is a great suggestion, thanks. We re-ran our weekly inversions using the EDF inventory scaled down to match the (lower) EPA inventory of Zhang et al. (2020), with total Permian emissions of 1.2 Tg  $a^{-1}$  instead of 2.7 Tg  $a^{-1}$ . We find that the Kalman filter is robust to this change after a burn-in period of 7-8 weeks. We discuss this new result in Section 3.1 and Section 3.5. This comment also spurred new discussion in Section 3.5 of the importance of spatial bias in prior emission estimates.

6. Figure 4 caption. Change "Figure" to "figure. Also, DOFS was already defined on Page 6, line 171.

Thank you for the correction. We include the definition of DOFS in the caption as an aid to readers who skim the figures.

7. Page 16, line 398: I think the paragraph should start with "Table 2 summarizes..."

Yes, that's right.

**Reviewer #2**

1. In general, there are a few published papers in the same region that used similar approach (i.e. Zhang et al., 2020), although this submission is conducted at a weekly scale instead of a monthly or annual scale, the authors still need to address the improvement of their study from previous ones, i.e. approach in the inversion framework? or found the weekly relationship between CH4 emissions and other activity indexes? or used more robust

**observations? or prior emissions and background?**

We summarize the novelty of our study in the introductory paragraph: "Here we show how satellite observations can be used to quantify weekly temporal variability in oil and gas methane emissions from a major production basin (the Permian) over a ~30-month period, and we show that this temporal variability can be explained by specific activity drivers" (L. 48-51). We also summarize how this differs from previous contributions: "A number of studies have used TROPOMI observations for inverse analyses of emissions at the scale of individual oil and gas basins (Zhang et al., 2020; Schneising et al., 2020; Liu et al., 2021; Shen et al., 2021; 2022; McNorton et al., 2022), but have generally focused on seasonal or annual estimates" (L. 56-59).

2. Line 19-21 "The mean oil and gas emission from the region (± standard deviation of weekly estimates) was 3.7 ± 0.9 Tg higher than previous TROPOMI inversion estimates that may have used too-low prior emissions or biased background assumptions". It seems the inversion results are sensitive to prior emissions, have you tested or quantified this potential bias of using different prior emissions to your results?

Great question, thanks. See our response to Reviewer #1, comment #5. We ran an additional weekly inversion using a down-scaled version of the EDF inventory as initial prior estimate, following the 2012 EPA inventory (1.2 Tg a-1 instead of 2.7 Tg a-1). These results are now discussed in Sections 3.1 and 3.5.

3. Line 88-89, "19346 ± 13073 observations per week over our full inversion domain (96°-110°W, 25°-38°N), including 3062 ± 2314", The standard deviation of available data numbers in each week is so high and is comparable with averages, which indicates there are not enough data in some weeks, please address what the potential bias for emission inversion in these weeks with lower available observation numbers. And can you display the time series of available CH4 observation numbers in each week?

Thanks for raising this point. We report that 124/127 of our weekly inversions have sufficient degrees of freedom for signal (DOFS) to estimate basin-wide emissions (L. 272). Inversions for weeks with few to no observations/DOFS return the prior. We now clarify that "Inversions with low DOFS are mainly constrained by the prior emission estimate" (L. 271-272). Figure 4 currently shows the number of observations and DOFS for each week; we do not feel that a time series of observation count merits an additional figure.

For some rainy or cloudy weeks, the available observation data can be sparse, leading to a large missing data gap in the study domain, and how will this situation affect inversion results for this large region? The reason to mention this comment is that your following analysis of the relationship between CH4 emissions and activity indexes ignored the influence of available data.

We find that we have sufficient observational information to infer emissions for the vast majority of our study period (124/127 weeks). To address your point that low information

content can introduce error to the interpretation of temporal trends, we now say that inversions with low DOFS "may introduce smoothing error to the inference of weekly temporal trends" (L. 272-273).

4. Line 92-94, "We use dynamic 3-hour boundary conditions from a global 4°×5° simulation corrected with spatially and temporally smoothed TROPOMI data as described by Shen et al. (2021). A one-month spin-up simulation starting from these boundary conditions is used for initialization". As we know that CH4 background uncertainty (bias) will be carried on to calculated CH4 enhancement, which is directly related to posterior CH4 emissions, what the bias of CH4 background in this study and potential uncertainty in deriving CH4 emissions?

As we say in the passage quoted here, our background is corrected relative to TROPOMI to remove bias between GEOS-Chem and the observations. Some spatially variable bias may remain due to model representation and smoothing error on the  $4^{\circ}\times5^{\circ}$  grid, and we include 8 coarse buffer elements in the state vector to mitigate this (discussed in Sect. 2.1, 2.2, and 3.5). We further discuss background/boundary conditions as a source of error in Section 3.5 (L. 375-399). Running an ensemble of inversions with different boundary condition datasets is beyond the scope of this study.

5. Line 110, "It attributes 94% of Permian emissions to oil and gas activity, and we assume the same fraction for our posterior emission estimates.", Can you clarify what is the potential uncertainty of using the constant fraction of 94% to oil and gas activity in inversion results? As I know most inversion studies have the ability to constrain posterior emissions from different categories.

Emissions in the Permian basin are dominated by oil and gas, which inversions cannot typically distinguish due to the spatial correlation of those sectors. Assuming a fixed prior sectoral breakdown of emissions (e.g., Z. Chen et al., 2022 *ACP*) is common in satellite-based estimation of regional methane emissions. There is some error in the sectoral breakdown of the EDF inventory, but in any case the emissions are dominated by oil and gas, so this would be a small error. We now state this on L. 218-220.

6. Section 2.2, As displayed in the reference list, there are some other inversion studies in the same domain, (i.e. Zhang et al., 2020), it's better to illustrate what the main improvement of your study when compared with these previous studies, because it's very hard for audiences to remember and distinguish the method difference between all related studies.

**See response to comment #1.**

7. Line 125, "mitigate boundary-condition errors (Shen et al., 2021; Varon et al., 2022).", have you assessed the improvement of CH4 background with observations?

The boundary conditions are corrected relative to TROPOMI observations. We find that our inversion (with 8 coarse buffer elements in the state vector) improves the mean

TROPOMI bias in the Permian basin from -9.0 ppb to -2.0 ppb. We now state this in the text on L. 232-233.

8. Line 136-137, "The error covariance matrices and are assumed diagonal with uniform error standard deviations of 50% and 15 ppb, respectively", As I understand, the 50% uncertainty for prior inventory may represent regional averages not all grid cells in study domain, which can be much larger than 50%, the same as 15 ppb for observations and GEOS-Chem simulations, so whether the inversion results are sensitive to different values of 50% and 15 ppb, if you assign a slightly larger extent (i.e. 80%, and 20 ppb), how much will the results change?

Thanks for this question. Our sensitivity inversions vary  $\gamma$ , which has the same effect on the cost function as varying the prior and/or observational errors. Our assumption of uncorrelated errors exaggerates the information content of the observations, and  $\gamma$  corrects for this. We now clarify this in Section 2.2 (L. 149-152).

In figure 3, the weekly emission changes can vary by 100%, indicating the potential bias or uncertainty of CH4 emissions at weekly scale can be much larger than 50%.

That's true. The 50% error on individual state vector elements is an error standard deviation (L. 148), so larger deviations are allowed. We now clarify that our assumption of uncorrelated prior errors would underestimate the error on regional aggregated emissions (L.150-151). Our sensitivity inversions varying  $\gamma$  better quantify uncertainty on the regional scale (L. 152-154).

**9.** Line 190, It seems the use of proportion 94% will largely influence your results of CH4 emissions from oil and gas. I am curious why the inversion model cannot constrain CH4 emissions from each category?

See our response to comment #5.

10. Data displayed in Figure 5 for model simulated xCH4 and observation.

Overall, why tower based CH4 concentrations seem higher than simulations with both prior and posterior emissions (scatter plot is below 1:1 line)? Whether it indicates the posterior CH4 emissions are still underestimated? How about plotting time series of concentration?

Whether one of the reasons for the large difference between model simulation and tower observations is the vertical gradient in the lowest GEOS-Chem model? And what is the height of the lowest model level? or aggregation error for spatial resolution between the point scale and regional scale(25km)?

Line 256-257, "The mean satellite-inferred emission (0.72 Tg is 20% lower than the mean tower (0.88 Tg and Scientific Aviation (0.89 Tg estimates during the period of overlap", From the above concentration comparisons, I also agree that the satelliteinferred emissions are obviously underestimated.

Thanks for raising this point. The good agreement with aerial emission estimates (Fig. 6) and higher basin-wide estimates than previously reported (Table 1) would suggest that our satellite-inferred emissions are not significantly underestimated. More likely is that the GEOS-Chem error is due to representation error (including vertical, as you point out) and transport error on the model grid, which we mention on L. 308-310. As we write on L. 305, the tower-model comparison is done in the nearest (3D) model grid cell.

11. Line 398-410, It's better to display the comparisons between atmospheric inversions and multiple linear regression with figures instead of only using tables and numbers.

Thank you for this comment. It prompted us to revisit our regression analysis. We found that predicting the four-week moving-average emissions, rather than the raw weekly estimates, leads to much higher explanatory power. We updated the analysis and discussion accordingly – including addition of a third regression model in Table 2 and new text in Section 3.6 (L. 485-531). We feel the table adequately communicates the key results that new well development and gas price are the most important predictors in this study.

12. Line 438, "assuming 80% methane content for Permian", whether the assumption of using 80% is reasonable, and what the general extent of this value in the study region?

We cite Alvarez et al. (2018) for the 80% gas content figure, which is widely used in the literature (e.g., Zhang et al., 2020 *Sci. Adv.*; Shen et al., 2022 *ACP*).

**13.** Line 398, "summarizes the models and results" I just guess the first author forget to delete this sentence of comment from other co-authors.

Thanks, we meant to refer to Table 2 here. Fixed!

---

## Author Response (AR2)

**Responses to the editor**

We thank the editor for her comments and suggestions. Each comment is copied below in blue, and our responses follow in black. Line numbers refer to the tracked-changes document.

1. In line with reviewer #2, I stumbled over the sentence (lines 94/95): "On average we obtain (mean ± standard deviation) 19346 ± 13073 observations per week over our full inversion domain (96°–110°W, 25°–38°N), including 95 3062 ± 2314 per week within the Permian itself (Figure 1)." This kind of description suggests that the distribution follows a Gaussian shape which, however, is not the case. Fig. 4 shows that you have considerably more observations per week on the lower end. I suggest to express the distribution with other statistical quantifiers like the 5% and 95% percentiles.

   Thank you for catching this. We now express the distribution with the $5^{th}$ and $95^{th}$ percentiles (L.86-88).

2. Regarding Fig. 4, you replied to reviewer #2 (their item 3): "Figure 4 currently shows the number of observations and DOFS for each week; ....". I don't think this is correct. Currently, Fig. 4 shows how often a certain range of observation counts was made within one week. The reviewer's intention, however (as I understand), was to see how the inversion behaved in the case that several weeks with low observation numbers occurred. I cannot retrieve this information from Fig. 4 in its current state. I suggest that you, at least, describe if this happened (low observation numbers in several consecutive weeks), and how the inversion behaved in this case.

   We now report the number of weeks between the 3 inversions with low DOFS < 0.5 (45 and 7 weeks), and state that we did not observe these weeks consecutively (L. 240-241).

3. Related to this topic is your statement in lines 267 to 269: "The spread around the best-fit line is because the number of DOFS depends also on the absolute uncertainty of the prior emissions, which varies both spatially and temporally, and on the spatial sampling of observations across the inversion domain." I am not convinced that this is fully correct. I suspect that the smaller scatter for smaller observation counts is caused by stronger regularisation towards the a priori. Later you mention the smoothing error which is indeed helpful. However, you might want to re-consider your statement in lines 267-269.

   Great point, thank you. We removed the sentence in question.

4. Finally, I have a small technical comment: In Fig. 4, x-axes with scales are missing for the number of observations and DOFs.

   We added background grids to the histogram plots to clarify that they have the same axes as the adjacent scatter plot.

In addition to these changes, we made 6 more minor edits:

1. Updated the TROPOMI version retrieval number from "2.2.0" to "02.02.00" as reported in the TROPOMI user guide (L. 84)
2. Updated Figure 6 to include the fifth sensitivity inversion from Figure 3.
3. Updated the caption of Figure 9 to explain why we exclude the fifth sensitivity inversion from that plot (L. 513).
4. Specify that the EPA inventory used by Lu et al. (2023) is spatially biased (L. 518).
5. Added a competing interest statement that co-author Ilse Aben is on the ACP editorial board.
6. Updated the Lu et al. (2023) citation since that paper is now published.